

# Identifying water deficit and vegetation response during the 2009/10
# drought over North China: Implications for the South-to-North Water
# Diversion project
**Bowen Zhu, Xianhong Xie, Kang Zhang**
State Key Laboratory of Remote Sensing Science, School of Geography, Beijing Normal University,
Beijing 100875, China

18   Corresponding author: Xianhong Xie

20   E-mail: xianhong@bnu.edu.cn

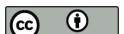



**Abstract.** Drought frequently occurs in North China and is the most damaging disaster in this region owing to its large-scale impact on hydrology and ecosystems. This is the main reason that China implemented the world-famous South-to-North Water Diversion (SNWD) project. However, quantifying the drought-induced water deficit at a regional scale is still a significant challenge. Gravity Recovery and Climate Experiment (GRACE) satellites monitor temporal variations in the Earth's gravitational potential and provide quality data sets for water storage analysis. In this study, we quantify the water deficit over North China in the context of the implementation of the SNWD project by focusing on a recent drought event, the 2009/10 drought, and identifying its onset, persistence, and recovery. As confirmed with ground-measured and land surface modelling data sets, GRACE can successfully capture temporal variations in total water storage. Total water storage shows a declining trend, reaching a low point during the 2009/10 drought with a water storage deficit of up to 25 km$^3$ (~22 mm). Groundwater storage shows a similar pattern, with a trend of −6.97 mm/yr. Together with the water deficit, vegetation growth is substantially restricted, as indicated by a reduction in the leaf area index. The amount of water transfer by the SNWD project can roughly meet the water deficit in North China but the effectiveness of the SNWD will depends on specific water configuration strategies.

Keywords: North China; drought; water deficit; Gravity Recovery and Climate Experiment; water storage



## 1 Introduction

The global climate system has significantly changed in recent years, leading to an increased frequency of extreme weather and other disaster events (Palmer, 2002). As a typical weather-related phenomenon, drought causes various problems such as the shortage of water resources (Lehner et al., 2006), crop damage (Deng, 2011), and ecological deterioration (Lewis, 2011), thereby imposing a direct threat to long-term security and social stability (R. Garcá-Herreraa, 2010;Jinsong Wang, 2012;Hsiang). Recently, drought has become one of the dominant factors limiting regional economic and social developments under the combined impacts of climate change and intensified human activities (Feng et al., 2014). With increasing water demand, population explosion, and uncertain water supply in the context of climate change, drought is expected to become more frequent and severe (Smith, 2013). Therefore, it is imperative to pay greater attention to drought events.

Drought frequently occurs in most areas of China and accounts for 35% of all economic losses from disasters. North China is an area with the most severe water shortage in China, particularly in arid and semi-arid regions (Feng et al., 2014); this area has shown significant sensitivity to drought events (Ju, 2006;Wei, 2003). To ease this situation, China has undertaken the South-to-North Water Diversion (SNWD) project to divert water from the Yangtze and Han Rivers from South to North China. The middle route of SNWD has been in service since December 2014 and provides water to hundreds of millions of

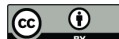



people on the North China Plain (NCP). Despite long-term planning and design of the SNWD project,
further demonstration and research is still needed to evaluate its actual resistance to drought.
During 2009/2010, a mega drought swept across the North China, causing a serious water shortage in
industry and agriculture as well as restrictions on vegetation growth (Barriopedro et al., 2012). A few
studies have focused on the drought in terms of meteorology, ecology, and economy. Gao and Yang
(2009) indicated that the La Niña event of 2008–2009 increased the differences in temperature and
atmospheric pressure between the Indo-Pacific Oceans and the Asian continent, causing severe
winter-time droughts in northern China. The drought might have been the main driving force behind the
decreasing trend in vegetation activity in North China: the summer droughts in 2007 and 2009 reduced the
vegetation cover by more than 13% (Wu et al., 2014). Moreover, the drought led to price fluctuation of
agricultural products in North China, despite the minor impact on main agricultural products (Lin et al.,

66  2013).

However, few of these studies have studied this drought event from the hydrological perspective. The
state of water storage in an area of interest is a direct hydrological response to the degree of drought, and
water storage anomalies can affect the hydrological cycle (Li et al., 2012). Regional-scale water storage
can be well quantified using data from the Gravity Recovery and Climate Experiment (GRACE). The
GRACE data have been successfully applied for water resources analysis in many areas such as central
North America (Wang et al., 2012) and North China (Feng et al., 2013).





In this study, we aim to explore the drought condition of North China during the past decade,
especially focusing on the 2009/10 drought, and to discuss whether GRACE can capture the typical
drought in North China. Moreover, we roughly evaluate the amount of water transferred by the SNWD in
remediation of the drought.
This paper is organized as follows: Section 2 describes the study area, data sets, and methods. Section
3 presents the results for SPI values and temporal and spatial changes in water storage. Groundwater and
surface water changes are also described. Sections 4 and 5 list the discussions and draw conclusions,
respectively.
**2 Data and Methods**
**2.1 Study area**
The region of interest in this study is North China (Fig. 1), which frequently experiences drought
events. North China covers an area of about 1.16 million km$^2$, is located in the region between 35–45 °N
and 110–125 °E, and has a climate predominantly influenced by the Asian monsoon. This region is in a
semi-arid environment with annual precipitation of around 500 mm (Fig. 1(a)), with most precipitation
occurring in summer; annual relative humidity of 53.6 %; and wind speed of 2.9 m s$^{-1}$ (Feng, 2012).
North China is an important area of grain production (Barriopedro et al., 2012); the main land cover



(39.5%) is cropland, with 33.6% grassland and 18.1% forest. Agricultural irrigation in the region is
heavily reliant on groundwater (Yang, 2010).

91       The topography of North China includes plains, mountains, and plateaus, with a declining slope from

northwest to southeast (Fig. 1(b)). The Inner Mongolian Plateau and the Tai-hang Mountain lie in the
north and west of the area; the NCP is in the center and southeast. The area contains drought-prone basins,
i.e., the Hai River basin and part of the Yellow River basin (Qin et al., 2015). Due to the large population
(~168 million), the average per capita water resource is only 23% of the Chinese average. In the NCP,
more than 70% of fresh water comes from groundwater (Zheng et al., 2010), which means that
groundwater plays an important role in local normal life, agriculture, and industry. Because of the uneven
spatial–temporal distribution of water resources, the economic losses and ecological disruption caused by
drought events can be more severe than in other regions.
**2.2 Data sets**
**2.2.1 GRACE data**

102       The GRACE satellite mission was launched by the National Aeronautics and Space Administration

(NASA) and the German Aerospace Center in March 2002. The GRACE project monitors temporal
variations in the Earth's gravitational potential. After atmospheric and oceanic effects have been
accounted for, the remaining signal on monthly to inter-annual timescales is mostly related to variations



in terrestrial water storage (Landerer and Swenson, 2012). Although its spatial resolution (~160,000 km$^2$)
and temporal resolution (ten-day to monthly) are low in comparison with other satellites, GRACE has the
attractive advantage that it senses water stored at all levels, including groundwater (Rodell et al., 2009).
Many studies have evaluated the use of GRACE satellites to monitor the hydrologic impacts of droughts
(Long et al., 2013) and long-term total water changes.

111        The GRACE data used in this study were processed by the University of Texas Center for Space

Research (CSR) using a Gaussian filter with a 300km smoothing radius to remove the stripes observed in
the spherical harmonic coefficient fields (Swenson, 2006). Data from the German Research Centre for
Geosciences (GFZ) and the NASA Jet Propulsion Laboratory (JPL) (http://grace.jpl.nasa.gov/data/) were
also used. Atmospheric and oceanic circulations had already been removed from mass distributions, and a
correction had been made (Rasums Houborg, 2010). Our GRACE time series included 120 approximately
monthly data points from January 2003 to December 2012. Anomalous fields were obtained by
subtracting out the multi-year mean field and converted to equivalent water heights including changes
regarding surface water, soil moisture, and groundwater, with a spatial resolution of 1°. We also isolated
groundwater changes by distracting the soil moisture and canopy storage changes from the total water
anomalies (Castle et al., 2014) to compare with the groundwater water change (GWC).



### 122 **2.2.2 Simulation data**

To diagnose the dryness of the 2009/10 drought and to validate the terrestrial water storage
measurements of GRACE, water fluxes (i.e., runoff and evapotranspiration) and soil moisture from two
land surface models were used in this study. The first is the Variable Infiltration Capacity (VIC) model
(Liang, 1994). VIC is a semi-distributed macroscale hydrologic model which solves full water and
energy balances. A number of improvements have been made to VIC so that it can deal with
complicated hydrological processes. Besides natural hydrological processes, VIC can consider water
management impacts associated with reservoir operations, and sprinkle irrigation (Haddeland et al.,
2006;Haddeland et al., 2007). The model's meteorological driving data mainly include precipitation,
wind speed and air temperature. The VIC model has been widely applied to analyze drought events at
regional and global scales (Andreadis, 2005;Sheffield and Wood, 2007;Xie et al., 2015). In this study,
The VIC daily simulation data at 0.25-degree resolution were obtained from Zhang et al. (2014) which
produced a long-term hydrological dataset for China specially. The model has been successfully
calibrated and validated using ground-measured streamflow and soil moisture, and remote-sensing
evapotranspiration (Zhang et al., 2014).
To perform a more extensive examination, we also used the simulated hydrological data from the
Global Land Data Assimilation System (GLDAS; (Rodell et al., 2004)), which incorporates four land
hydrological models (LSM, CLM, VIC, and NOAH). The NOAH model has more than 30-year history



(Chen et al., 1996). The model is driven by near-surface atmospheric forcing data including air
temperature, air humidity, and precipitation (Charusombat et al., 2012). It simulates surface water and
energy balances such as soil moisture, soil temperature, canopy content, and water and energy flux terms
(Yang et al., 2013). The NOAH model has undergone continuous improvement (Ingwersen et al., 2011),
and it has been included in the GLDAS in which ground-based and space-based observations were used
to estimate the land surface states (Fang et al., 2009). To verify the GRACE measurements, in this study,
we used the NOAH simulated data from GLDAS because the data were widely applied (Rodell et al.,
2009;Long et al., 2013;Syed et al., 2008) and they have also been evaluated in North China with
acceptable uncertainties (Feng et al., 2013;Huang et al., 2015).
Please note the VIC and the NOAH simulation data of water fluxes and soil moisture were from other
studies, and we did not perform the simulations. Their daily data at 0.25-degree resolution were
aggregated to monthly and one-degree scale to compare with GRACE.
**2.2.3 Ground-based measurements and other data**
In this study, ground-based measurements of precipitation, groundwater, and surface water storage
were used. Ground-based measured precipitation data from the Chinese Meteorological Administration
were applied to derive gridded precipitation at a spatial resolution of 0.25 ° using the synergraphic
mapping system algorithm (Shepard, 1984). The gridded precipitation data have been extensively verified



for runoff, evapotranspiration, and soil moisture (Zhang et al., 2014). These gridded precipitation data can
be used to identify the spatial coverage of meteorological droughts.
In order to detect the impact of the drought on the groundwater system, groundwater table
observations were acquired from 95 observation wells. The distribution of these wells is shown in Fig.
1(b). Reservoir storage constitutes a major part of surface water, so water stored in reservoirs in the Hai
River basin in 2003–2012 Hai River Water Resources Bulletin (HRWRB) were also used to examine this
drought. Moreover, the data of annual groundwater withdraw from the HRWRB were applied to reflect
the human activity on groundwater storage.
**2.3 Methods**
We first characterized the 2009/10 drought in a long perspective based on the 53-year precipitation.
The Standardized Precipitation Index (SPI) and the probability of yearly precipitation are used to
represent the status of the drought in the 53 years. Then we identify the water storage condition, including
the total water storage, surface water and groundwater. In order to evaluate the GRACE data, we
compared net recharge from GRACE and the simulated data. Moreover, the groundwater storage
calculated from GRACE was also evaluated using in-situ observations. Here we specially present the
methods used to calculate the SPI, net recharge, and groundwater storage.



### 2.3.1 SPI

The severity of a drought can be quantified with a drought index. The SPI was used to reflect the

meteorological drought, which was proposed by McKee (1993) and is a widely used drought index. The

index is a statistical monthly indicator that compares the accumulated precipitation during a period of

specific months with the long-term cumulative rainfall distribution for an accumulated period (Nam et al.,

2015). The timescales of SPI vary from 1 month to 24 months. When the time periods are small (1 or 6

months), the SPI frequently fluctuates above and below zero (McKee, 1993). In this study, 53-year

monthly precipitation data were used to calculate the SPI, thereby diagnosing the severity of the

2009/10 drought.

### 2.3.2 Net recharge of total water storage

As the same to many satellite data, uncertainties in GRACE are inevitable caused by atmosphere,

sensor and other factors. The GRACE data need evaluation for the area of interest. Therefore, we

calculated the monthly net recharge of total water storage ($\Delta S$) from two independent sources: the model

simulations (i.e., from NOAH and VIC) and the GRACE data (Famiglietti et al., 2011). As the GRACE

monthly data represent the mass anomaly, the difference of the GRACE data in two successive months is

equivalent to the monthly net recharge (Wang et al., 2014):

$$\Delta S_i = S_i - S_{i-1} \qquad (1)$$





where the subscript *i* stands for the *i*th month and $S_i$ represents the *i*th month total water storage anomaly.
With the model simulation data (from NOAH and VIC), the net recharge can be computed based on
the monthly basin-scale water balance (Syed et al., 2008):
$$\Delta S_i = P_i - E_i - R_i \tag{2}$$
where *P*, *E*, and *R* denote precipitation, evapotranspiration, and runoff, respectively.
Therefore, the agreement of net recharge calculated from Eqs. (1) and (2) is a useful indicator for the
accuracy of GRACE in capturing the total water storage change, because the model simulation and
GRACE are independent approaches (Syed et al., 2008).

### 2.3.3 Groundwater storage

Groundwater is an important part in the total water storage in North China. To detect groundwater
changes during recent years, the storage variation is discussed. There are two methods for calculation of
groundwater storage change (GWC). The first method is based on ground measurement by multiplying
the measured groundwater level anomalies by the specific yield of each well (Huang et al., 2015):
$$G_i = H_i \cdot \mu \tag{3}$$
where $H_i$ represents the groundwater level measured in situ for the *i*th month and $\mu$ stands for the specific
yield. In this study, the value of $\mu$ for each site was prescribed based on the soil properties according to
Huang et al. (2015).





The other method for GWC computation is subtraction of soil water storage from the GRACE total
water storage changes:
$G_i = S_i - M_i - C_i - W_i$           (4)
Where $G$ is the GWC, $S$ and $M$ denote the GRACE total water anomalies and the soil moisture changes
simulated by the hydrologic model, respectively. The $C$ and $W$ represent canopy water storage and
surface water (i.e., water storage in reservoirs), respectively.
Through the two methods, groundwater storage is obtained so that to evaluate the GRACE data
and to quantify groundwater changes.

## 3 Results

### 3.1 Precipitation deficit

Precipitation is a direct indicator of drought. We used monthly precipitation data to analyze the water
balance input during 2009 and 2010 (Fig. 2) and diagnosed the dryness. As illustrated in Fig. 2, the
regional average accumulated precipitation is less than the climatological mean values calculated for the
period 1960–2012. Especially in the summer and the fall of 2009, the precipitation only accounts for 78%
of the climatologically mean. The spring of 2010 is slightly wet due to a near-normal monsoon season
(Barriopedro et al., 2012). The regional precipitation deficit reaches 14 mm throughout 2009/10 and 47
mm from May 2009 to April 2010.



To characterize this drought well, 53-year monthly precipitation data (from 1960 to 2012) were used
to calculate the SPI. Three timescales of SPI are shown in Fig. 3(a), indicating different drought situations.
Meteorological and soil moisture conditions respond to precipitation anomalies on relatively short
timescales, whereas streamflow, reservoirs, and groundwater respond to long-term precipitation
anomalies on the order of 6 to 24 months or longer. According to the SPI classification (Nam et al.,
2015;Qin et al., 2015), the 12-month SPI (approximately −1.0) indicates a moderate drought during May
2009 to April 2010, the 1-month SPI represents a severe drought in August and October 2009, and the
6-month SPI indicates a severe drought from October to December 2009 with the lowest SPI value of
approximately −1.63. Overall, there is an obvious drought event in North China from May 2009 to April

233    2010.

In addition to the SPI, the probability of yearly precipitation can also reflect the water input
conditions with respect to North China. To compute the probability, we first defined the hydrological year
as being the period between this May and the next April. We sorted the 52 years of precipitation from high
to low and calculated the probability of each year using the Weibull equation (Helsel D, 2002). Figure 3(b)
shows the results: the precipitation of 2009 was ranked 43rd, and the probability of precipitation during
this drought period was only about 84%, indicating that 2009 was a severely dry episode during the 52
years, which is consistent with the SPI results.

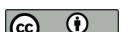



## 3.2 Total water storage


The lack of water input (i.e., precipitation) during the drought period probably induces a decrease in
water storage. As shown in Fig. 4(a), the GRACE data from CSR, JPL, and GFZ have similar trends and
match quite well. Overall, there is a notable decrease of total water storage in North China from 2003 to
2013, indicating recurrence of the drought. The total water storage anomalies in 2009 and 2010 are below
zero with a mean value of approximately −21 mm and a minimum value of −40 mm, which means that
water storage is less than normal. The storage shows a small increase in the winter of 2009 and spring of
2010: this trend is consistent with the precipitation change.
There will be uncertainties in the GRACE data, so we verified the data by comparing with the net
recharge of water storage (ΔS) from the NOAH and VIC simulations. To make the comparison, the
average GRACE values from CSR, JPL, and GFZ were computed. From Fig. 4(b), the ΔS series of
GRACE agrees well with the values from VIC and NOAH, although ΔS of GRACE displays larger
fluctuations. The correlation coefficient between GRACE and NOAH is 0.53 and the correlation of
GRACE with VIC is 0.52, whereas the correlation between VIC and NOAH is about 0.85, suggesting a
certain degree of consistency between the three sources of data.
The spatial distributions of total water storage anomalies for this drought event are presented in Fig. 5.
From May 2009 to April 2010, the south of the region that contains Shanxi, Shandong, and Hebei
provinces suffered a much more severe drought than the north, especially in the summer and fall of 2009



and spring of 2010. Although the spatial distribution is uneven, total water storage is still below zero and
the south of North China is the main affected area.

261        Furthermore, we computed the relative departure of water storage for 2009/10 from the average.

From Fig. 6, we can see that drought events mainly occur in the south of North China, where the water
resources are very poor. The regional average water storage deficits are up to 22 mm, about 25.5 $km^3$
relative to the normal water storage condition.
**3.3 Response of surface water and groundwater**
**3.3.1 Surface water storage**

267        Due to data availability, data for yearly reservoir storage were used to reflect surface water storage.

According to *Water Resources Bulletin of Hai River Basin* (http://www.hwcc.gov.cn/), the number of
reservoirs slightly increased from 137 in 2003 to 146 in 2012, so the total water storage of reservoirs
increased from 61.1 $km^3$ in 2003 to 95.81 $km^3$ in 2012 (Fig. 7). To derive the surface water storage
changes, we use the average storage of the reservoirs. Long-term average water storage is about 0.16 mm,
but the storage reaches its lowest levels in 2009 (~0.13 mm) and 2010 (~0.14 mm), reflecting the
influence of the drought.



### 3.3.2 Groundwater change


Groundwater is a vital source of fresh water for agriculture, industry, public supply, and ecosystems
in North China (Feng et al., 2013). To quantify the influence of droughts on groundwater storage, in
addition to the GRACE data, we used the ground observations from the 95 wells. Figure 8(a) presents the
average variations of groundwater tables of the 95 wells. There is a gradual decline of approximately
−0.41 m/yr, despite substantial uncertainties. For the 95 wells, the trends in the groundwater table range
from −2.5 to 2.0 m/yr, and the decreases are mainly apparent in the south of North China (Fig. 8(b)).
Figure 9 shows the groundwater storage change derived from the in situ observations and GRACE, and
groundwater storage is described as the equivalent water height. Both of these data sets indicate a
downward trend, of 4.68 mm/yr for GRACE and 6.97 mm/yr for ground observations. This difference
may be attributable to the uncertainties within GRACE and ground observations and the spatial
representation of the 95 ground observations. Despite such differences, the changes in groundwater
storage from GRACE and ground observations have a strong correlation, with a Pearson correlation
coefficient of approximately 0.71.



## 4 Discussion

### 4.1 Further evidence and impact of the drought

Climate change in North China during past decades can be characterized as an increase in air temperature and a decrease in precipitation (Ming et al., 2015). Moreover, the frequency and intensity of drought over North China has significantly increased during the last five decades (Qin et al., 2015), mainly caused by the dramatic decrease in precipitation (Xu et al., 2015). In this study, we focus on the 2009/10 drought event in the context of the environmental changes in the past decade. Given the SPI values and the probability of precipitation, this drought was a severe event. The drought started in May 2009 and ended in April 2010, as shown by Barriopedro et al. (2012). In contrast to existing studies focusing on the drought from the viewpoint of meteorology or ecology, we addressed this drought event from a hydrological perspective in order to analyze the influence on water storage, which is essential for ecosystem and agricultural production.

With decreasing precipitation, water storage depletion has taken place during the past decade in North China (Moiwo, 2013). In this study, we found that surface water storage reached a low level in 2009 and 2010. The responsiveness of the groundwater system is important for hydrological drought development (Van Loon and Laaha, 2015). The groundwater table has displayed a continuous decline at a rate of ~0.3 m/yr since 1960 (Cao et al., 2016).





One may wonder the role of human over-use of the water resources. Figure 10 shows total
groundwater withdraws for 2003-2013. Although the groundwater withdraws continuously decreased
during the past decade, it primarily contributed to the groundwater decline in North China, because there
is no significant trend in the net recharge (Fig 4b). Similar results were also shown in Zheng et al. (2010).
However, the water deficit during the 2009/10 drought is dominated by the inadequate precipitation input,
so that the groundwater storage is at the low level during the period (Fig 9). Moreover, our study shows
that the rate of groundwater decline is approximately 0.41 m/yr from 2005 to 2014, indicating an
accelerating depletion, which may be attributable to the reoccurrence of drought events.
**4.2 Impact on vegetation**
In addition to the water storage depletion, the typical 2009/10 drought induced negative impacts on
vegetation growth (Wang et al., 2015;Zhang et al., 2016). Wu et al. (2014) indicated that this drought
probably reduced the normalized difference vegetation index by 6.68% in 2009 in the Beijing–Tianjin
sand source region.
To investigate the impact of this drought further, we calculated the average leaf area index (LAI)
within the growing season (from May to October) for three types of land cover (grass, crop and forest), as
LAI is an important indicator of crop growth and plant productivity (Liang et al., 2015). As shown in Fig.
11, LAI reaches its lowest level during 2009. Especially for crop land, LAI in 2009 is less than its
multi-year mean of approximately 0.11. An area of more than 0.3 million km$^2$ of North China shows a



substantial LAI reduction. It should be noted that the spatial distribution of the LAI reduction (Fig. 11(b))
is approximately consistent with the area of water storage deficit (Fig. 6). Thus, this drought event has a
negative effect on vegetation growth, and especially causes the reduction of agricultural production.
**4.3 Implications for the SNWD project**
The SNWD project supplies water resources from the Yangtze River basin to North China, and it is
expected to transfer approximately 27.8 km$^3$ of water annually. In this study, we demonstrated that the
2009/10 drought was a severe episode with precipitation ranking 84%, and the water storage deficit is
about 22 mm (~25 km$^3$). Therefore, the SNWD project can probably replenish the water deficit at this
level of drought. Certainly, the efficiency of the SNWD in combating drought will depend on the water
configuration strategy (Dong et al., 2012). However, the amount of water transfer by the SNWD is not a
constant, it depends on precondition of water resource regions and requirement of receiving water
regions (Zhang et al., 2011). During the summer monsoon rainy season in South China, the SNWD is
expected to provide a large amount of water resources to replenish the surface water and groundwater
storage in North China when a drought event occurs. In combating droughts and relieving the stress of
water resources, moreover, the SNWD project requires additional evaluations of water quality regarding
surface and ground water and the effect on ecosystems (Tang et al., 2014;Zhu et al., 2008).



## 5. Conclusions

In this study, the hydrological effects of the 2009/10 drought in North China are discussed using multi-source data, including satellite data, ground measurements, and model simulations. On the basis of the precipitation data, the shortage of precipitation was 47 mm from May 2009 to April 2010: this event is regarded as a severe drought on the basis of the SPI value. Moreover, the probability of precipitation during this period was about 84% in the past 52 years, also indicating a notable drought event, consistent with the SPI analysis. There was a declining trend in total water storage for the past decade based on GRACE data, and the regional deficit of water storage was approximately 22 mm (~25 km$^3$) in 2009/10. The relatively dry area is located in the south of North China. Furthermore, both groundwater storage and total water storage decreased year by year, while the surface water reached its lowest level in 2009. Thus, this drought event has led to damaging hydrological effects as well as suppression of vegetation growth in North China. The SNWD project may ease the water storage deficit in North China for this level of drought intensity.

The GRACE data have attractive advantages for large-scale drought and flood-potential detection (Li et al., 2012;Rasums Houborg, 2010;Reager and Famiglietti, 2009). However, the effective spatial resolution of GRACE is about 150,000 km$^2$ at best (Swenson et al., 2006), so these data may not be suitable for small-scale issues. With the implementation of the SNWD project, moreover, there is a growing need for real-time drought monitoring and forecasting. Use of multi-source data, including



satellite data, ground measurement, and model simulations, is an effective strategy to quantify both

drought intensity and water deficits.

**Acknowledgements:**

This study was supported by grants from the National Natural Science Foundation of China (No. 41471019, 41331173) and the National High Technology Research and Development Program of China (No. 2013AA121200).

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




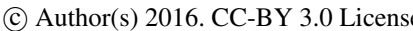


**Figure 1: (a) Location of North China (black line) and the Spatial Distribution of Annual Precipitation over China; (b) Topography and Distribution of Groundwater Gauge Stations (Red Triangles) in North China.**





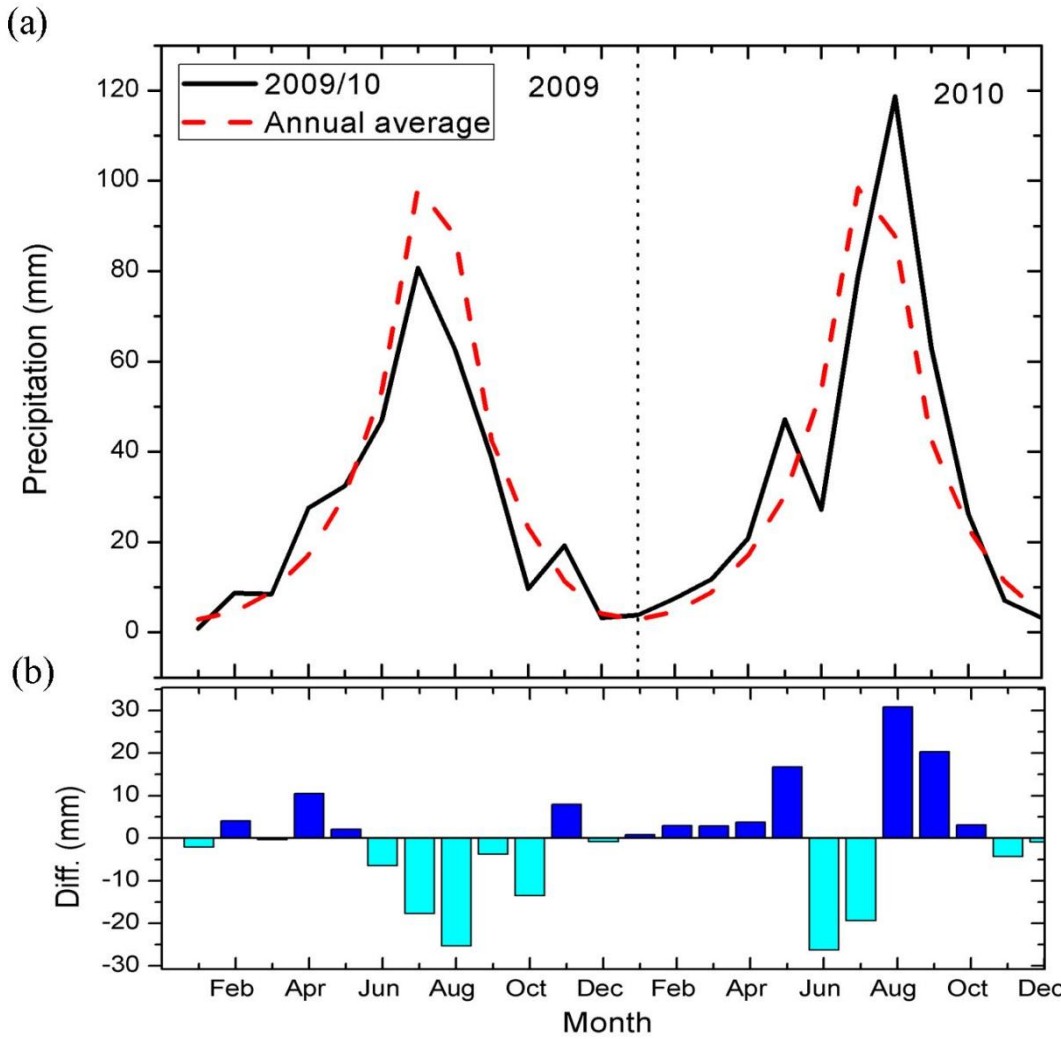


**Figure 2: Accumulated Monthly Precipitation during 2009/10 (a) Compared with the**

**Climatological Mean; (b) Monthly Departure from the Climatological Mean.**





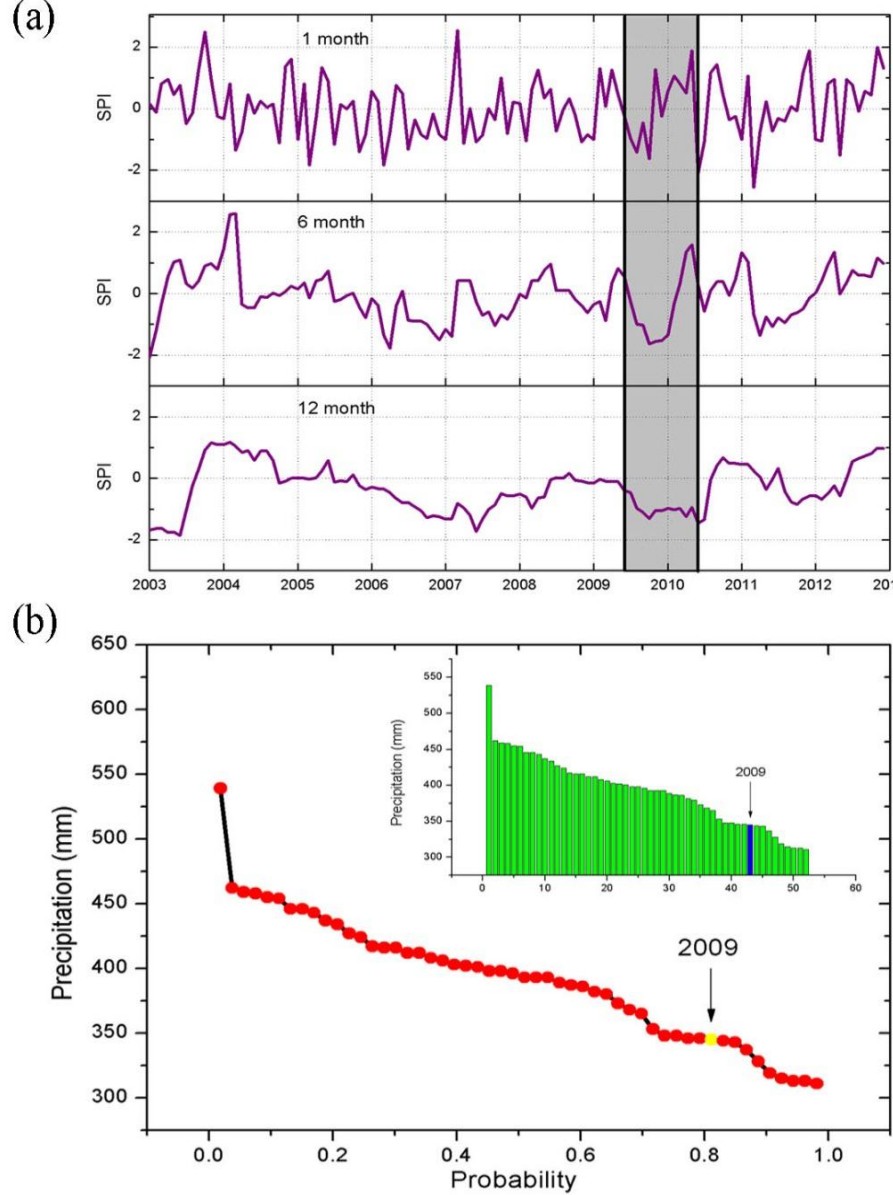


**Figure 3: (a) SPI on Three Timescales (1 Month, 6 Months, and 12 Months) for 2003–2012; (b)**

**Probability of the Hydrological Year's Precipitation. Green Bars are Annual Precipitation for**

**1960–2012.**



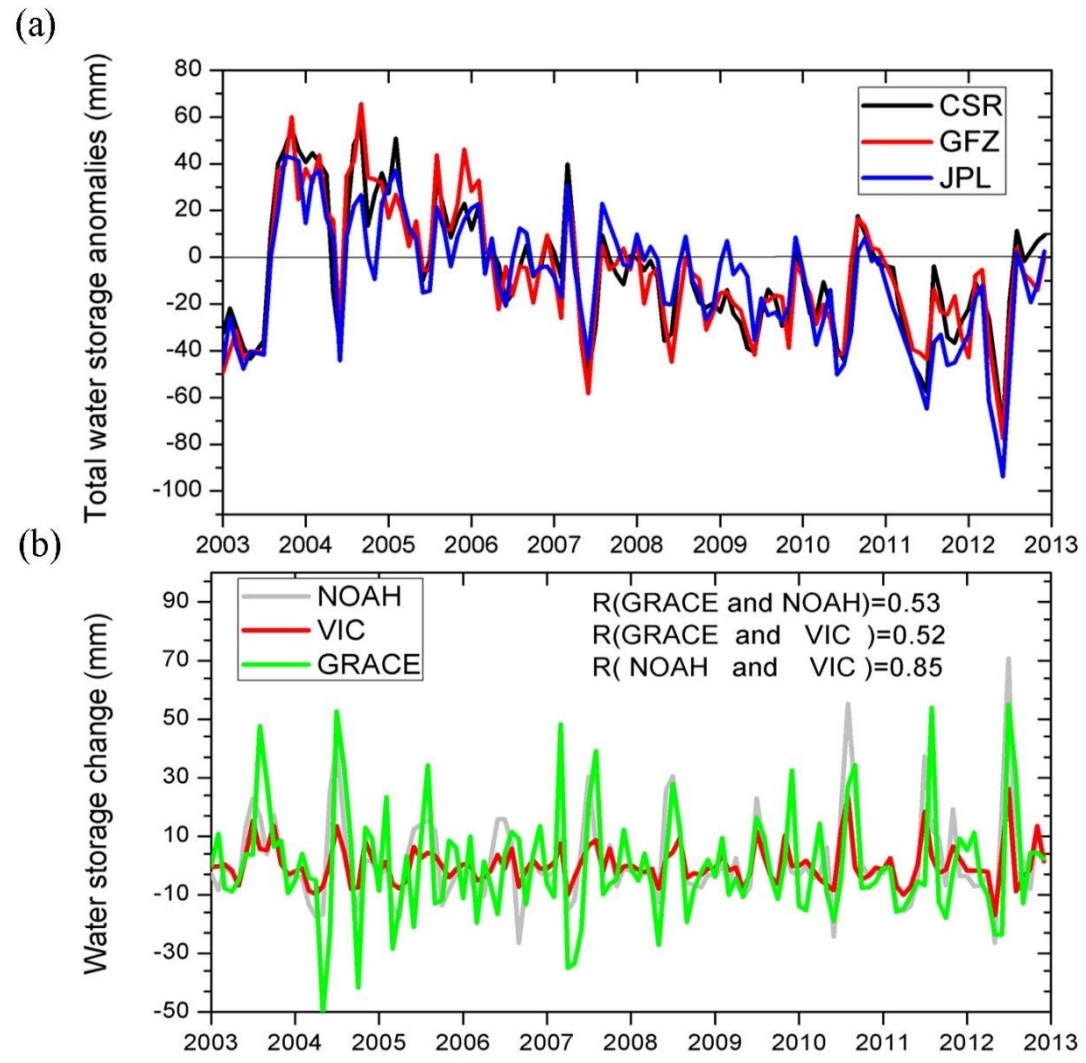


**Figure 4: (a) Total Water Storage Anomalies in North China from 2003 to 2013; (b) Comparison of**

**Three Models of Water Storage Changes.**





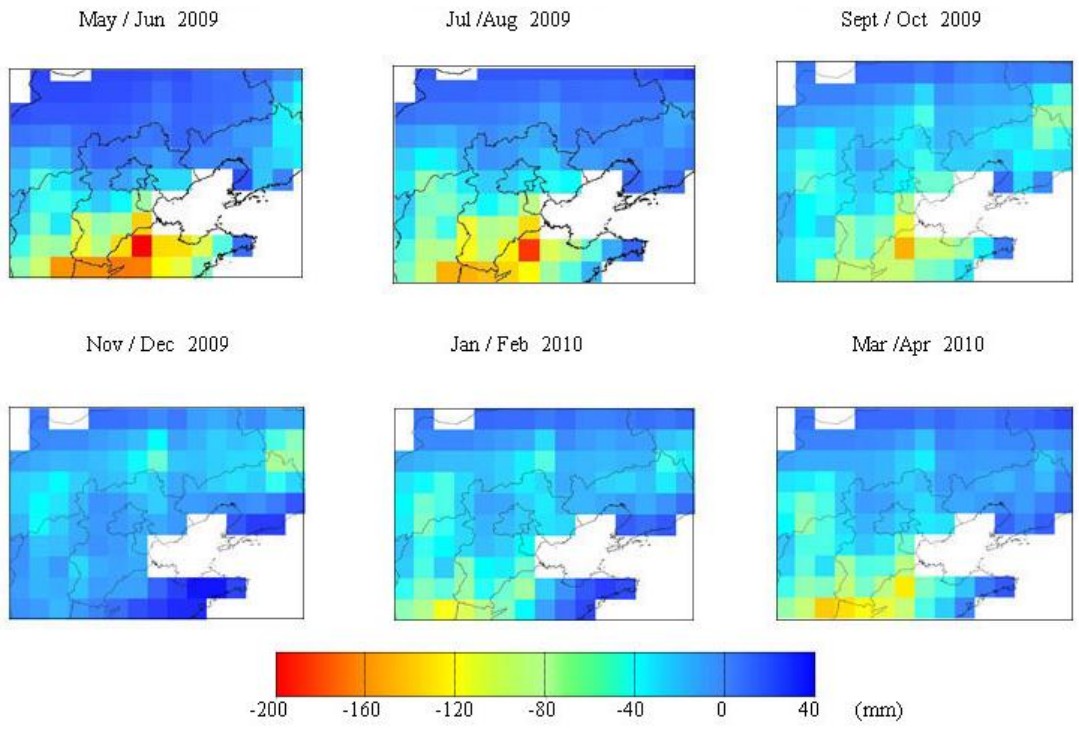

**Figure 5: Spatial Distributions of Water Storage Anomalies between May 2009 and April 2010.**





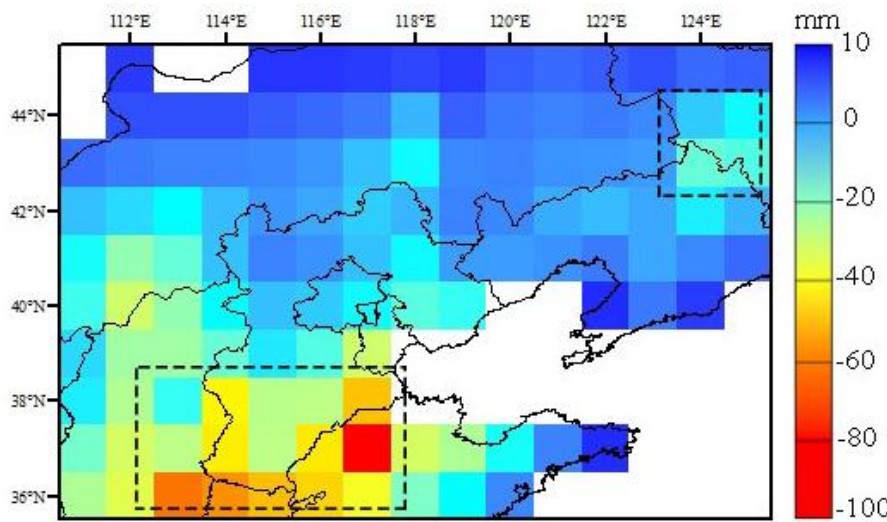


**Figure 6: Water Storage Deficits Relative to the Normal Water Storage Conditions from May 2009**

**to April 2010. The Dotted Line Shows the Seriously Dry Area.**













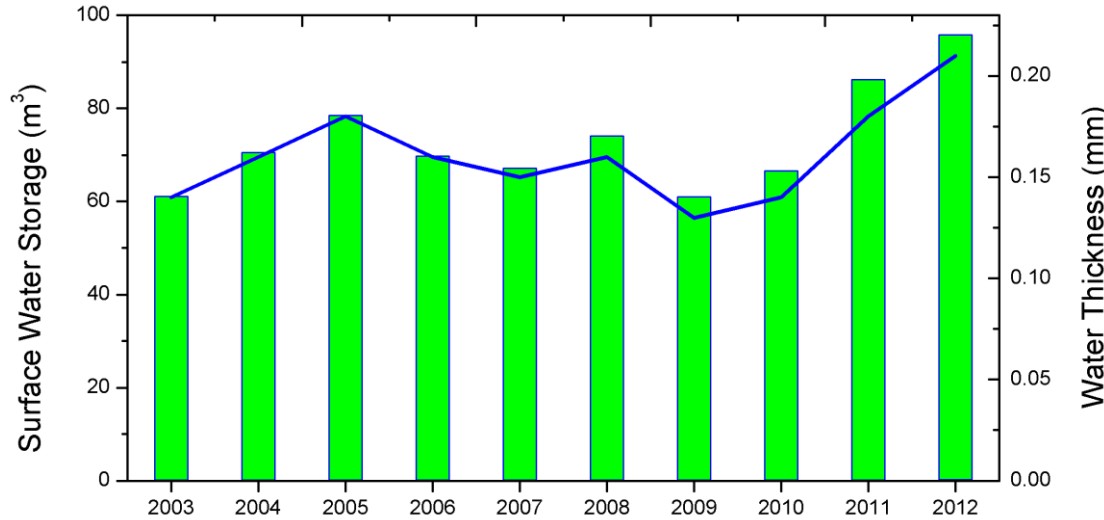


**Figure 7: Surface Water Storage (Green Bars) and Equivalent Water Thickness Changes (Blue**

**Line).**




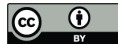

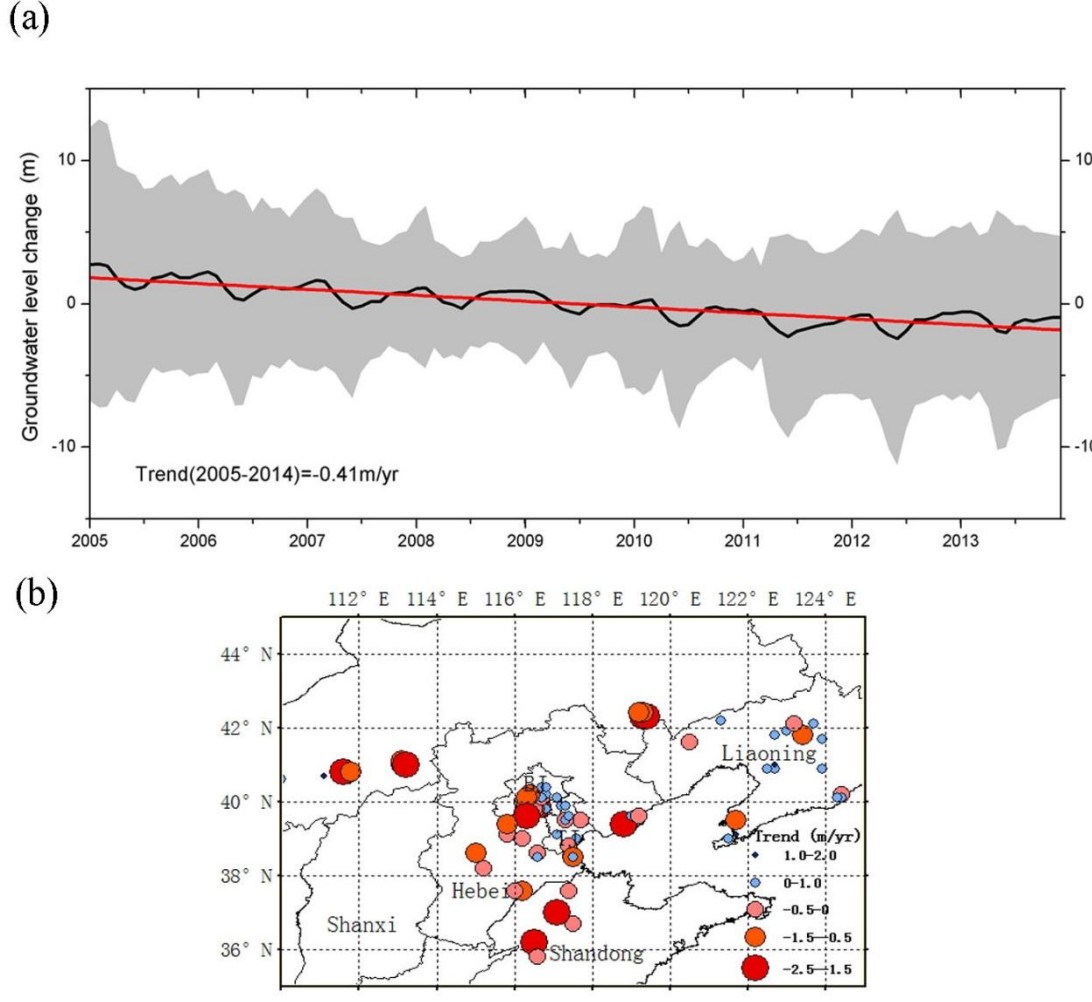

**Figure 8: (a) Groundwater Table Changes from 2005 to 2014 in North China. The Shaded Area Shows the Uncertainties (95% Confidence Intervals); (b) The Trend in the Groundwater Table for each Gauge.**



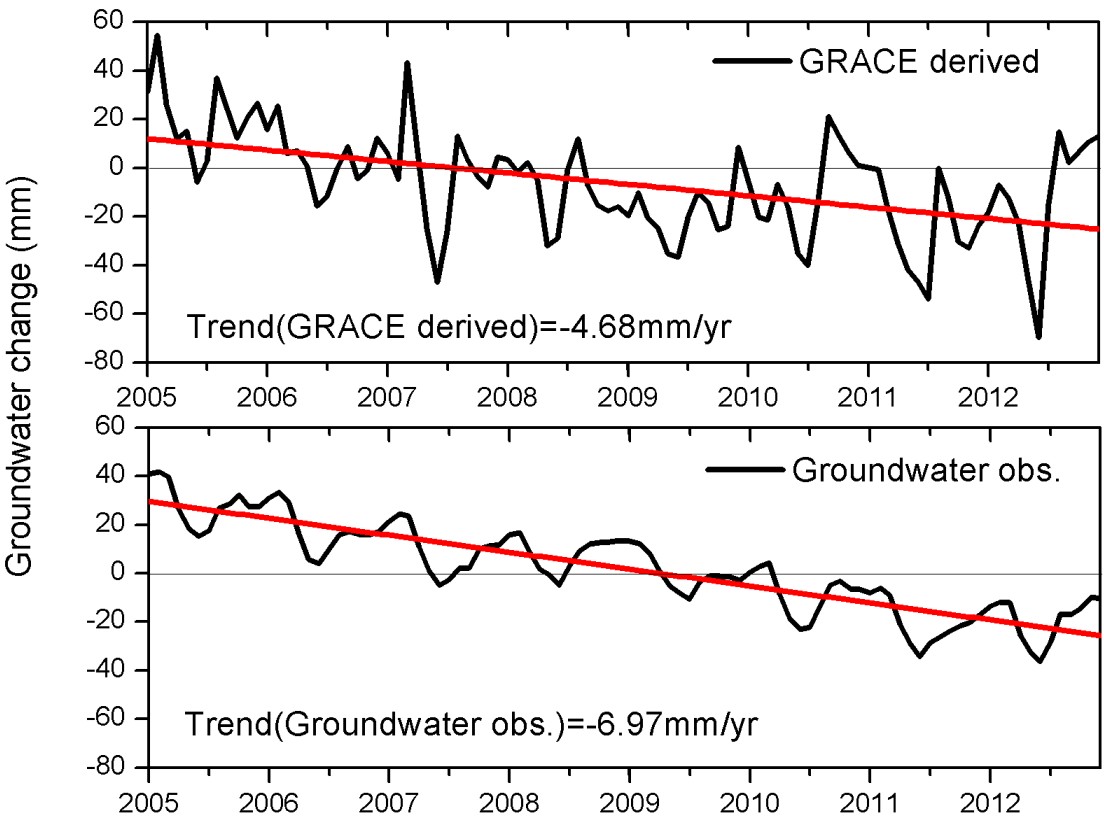


**Figure 9: Groundwater Storage Changes Derived from GRACE and Ground Observations.**









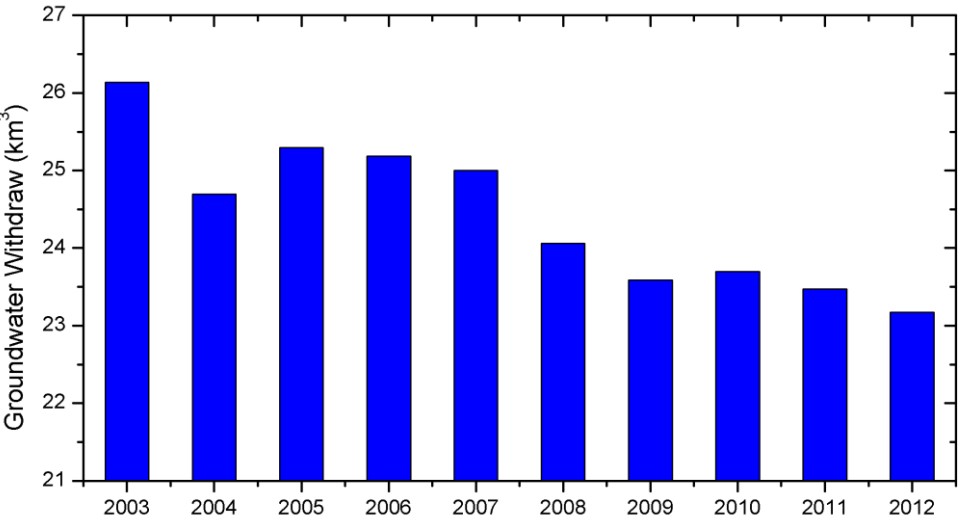


**Figure 10: Groundwater Withdraw Changes from 2003 to 2012 in Hai River Basin.**










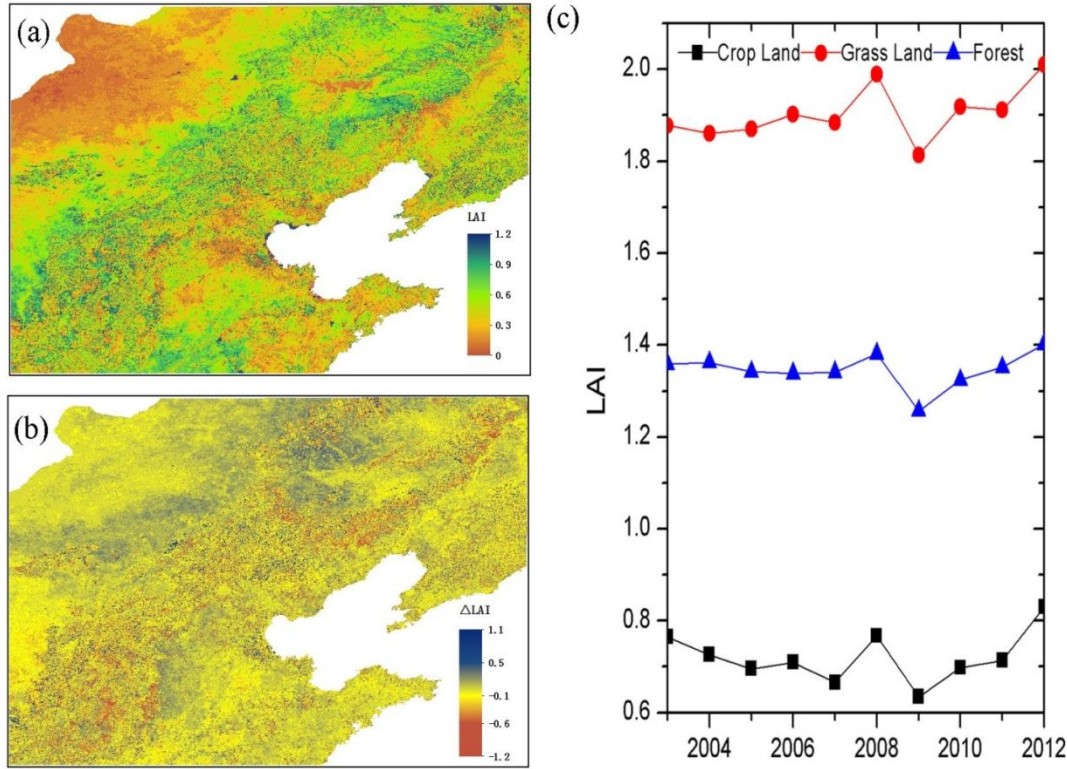


**Figure 11: Spatial and Temporal Distributions of LAI: (a) LAI for 2009; (b) Departure from 2009**

**LAI (2009 LAI Minus the Multi-year Mean); (c) Time Series of LAI Corresponding to Three Types**

**of Land Cover.**
