# Peer review of "Identifying water deficit and vegetation response during the 2009/10"

_Hydrology and Earth System Sciences, 2016_

## Referee Comment (RC1) · Anonymous Referee #1 · 6 Sep 2016

General Comments: This manuscript aims to identifying the water deficit under an extreme drought in North China by using the GRACE data, and comparing with the response of vegetation, towards the implications for the well-known South-to-North Water Diversion (SNWD) project. It is not new to investigating the GRACE-derived water storage changes under droughts. As the author mentioned that several previous studies have done this with varying focuses, but it remains very interesting in the North China, where the water shortage is aimed to be mitigated by the SNWD. Besides, due to the differences on the drought characteristics such as duration and severity, the detection and variation of GRACE-derived water storage changes may vary from place

to place. Thus, this manuscript addressed a good scientific question and may attract interests from the community of hydrology, geodesy, and even the public people. To achieve above purposes, the study derived the total water storage anomalies (TWSA) and groundwater storage anomalies (GWSA) from GRACE time-variable gravity data, and compared it with model simulations and LAI. The method used in this study is generally appropriate and the result is reliable. However, some efforts may still be needed to improve the quality of this manuscript and make it easier to be understood.

Major Comments: [1]Section 2.3.2, Evaluation of GRACE TWSC: This manuscript compares the so called net recharge (this name is confusing, actually it is commonly written as dS/dt or TWSC in many papers)derived from GRACE and land surface models. It should be noticed that in Equation (2), the runoff should be the net runoff (i.e. outflow minus inflow). Since the North China is a self-defined region with a shape of rectangle, it is definitely not a closed basin. How is the net runoff calculated? Please give more explanations. Instead of the net recharge, there is another way for evaluation, i.e. using the observations of groundwater storage (GWSA) and surface water storage (SWSA), and model simulations of the soil moisture storage (SMSA), with the equation: TWSA = GWSA + SWSA + SMSA. [2]Section 4.2, Vegetation response: More deep analysis is needed to figure out the vegetation response under the drought. For example, can the monthly LAI help to interpret the response while compared with the monthly TWSA?

Minor Comments: [1]Line 22: should be : is one of the most damaging. . . [2]Line 27: 'quality data sets' is confusing [3]Line 111: please point out the GRACE data used is Level-2 or Level-3 [4]Line 120: 'total water' should be 'total water storage' [5]Line 121: 'groundwater water change' should be 'groundwater storage change' [6]Line 159: 'groundwater table' is better to be replace with 'groundwater level' [7]Line 199: 'to detect groundwater' should be 'to detect groundwater storage' [8]Line 204: 'groundwater level measured in situ' should be 'in situ measured groundwater level' [9]Line 209: the symbol of G S M C W is not typically used in the GRACE hydrology community, I
suggest to using GWSA = TWSA – SMSA – SWSA – CWSA [10]Line 222: what is precipitation deficit? I can not understand how is the 14 mm and 47 mm derived. [11]Line 253: 'fluctuations' better to be replaced with 'amplitude' [12]Line 261: 'departure' is not easy to understand, usually we use 'anomaly' or 'difference' [13]Line 262: it is hard to say the drought events mainly occur in the south of North China, as the groundwater exploitation is complicated in space. [14]Line 264: 'normal' or 'average'? [15]Line 275: 'public supply' should be 'domestic use' [16]Line 283: 'downward' should be 'decreasing' [17]Line 287: delete 'approximately', same in many other places throughout the manuscript [18]Line 311: 'groundwater decline' should be 'groundwater level decline' [19]Line 352-354: references are not commonly found in the conclusion section [20]Figure 1: the year for annual precipitation, long-term mean or some specific year? [21]Figure 1: 'Groundwater Gauge Stations' should be 'Groundwater Level Monitoring Wells' [22]Figure 2(b): do not use abbreviation for the name of y-axis [23]Figure 2(a): what is the time for annual average, 53-year mean? [24]Figure 3(b): No name for the x-axis in the small figure inside. The green histogram and red dots seems represent the same thing. If not, please give more explanation. [25]Figure 5: What is the meaning of '/' in 'May/Jun 2009' and the others? Is it the average TWSA of May and June 2009? Please make it clearer. [26]Figure 7: The name of the right y-axis should be 'Equivalent Water Height (mm)'

---

## Referee Comment (RC2) · B.L. Finlayson (Referee) · 6 Sep 2016

General comment:

The focus on drought alleviation in relation to the SNWD seems inappropriate. Rather the SNWD was developed to deal with the high water demand in North China which is a problem irrespective of whether drought conditions occur. Further, over-extraction of groundwater from many areas of North China, especially the North China Plain, has a much greater and more permanent impact on groundwater storage than droughts. To ascribe North China's problems with water as being caused by droughts is an extreme

simplification of the water resources situation there.

The introduction is rather divorced from the content of the body of the paper which focuses on the use of GRACE in analysing water storage in the NCP region. The introduction should focus on the main topic of the paper with a review of key papers from which it can be shown that the work reported in this paper is a worthwhile development on what has been already done in this field and then go on to show how this will be achieved.

The Data and Methods Section (Section 2) lacks a clear description of the methodology of this study. It begins with a rather misleading description of the field area and describes the GRACE system and data sources, proceeding to say that the hydrological modelling described is taken from some other source and is not part of this research. The modelling is reported to have been "evaluated in North China with acceptable uncertainties" (lines 147-148) but with no description of what this means.

Water resources are heavily used in this region, especially in the southern part on the North China Plain, so the focus on drought and failure to take more account of resources use is strange. There is some discussion of this in the section from line 305 to 312, beginning: " One may wonder the role of human over-use of the water resources." Indeed.

Specific comments:

This paper begins with an incorrect statement by claiming that a paper by Palmer 2002 is the source for a statement that " The global climate system has significantly changed in recent years, leading to an increased frequency of extreme weather and other disaster events". The paper is actually by Palmer and Räisänen and it does not say that there has been a change in recent years leading to increased frequency of extreme events but rather it discusses the probability of this occurring in the future. This paper cannot be used to support such a claim for north China. Also, the paper was written 14 years ago and a lot more has been published on this topic since. I wonder how many

other references used in this paper would stand up to scrutiny?

Lines 49-50: "Drought frequently occurs in most areas of China and accounts for 35% of all economic losses from disasters." No source is given for this specific piece of information.

Line 78: 'SPI' not defined until line 167.

North China, as defined in Fig 1b, is not the area with the most severe water shortage in China (line 50); only the western part of that area is semi-arid; and based on the authors' own map of precipitation distribution (Fig 1a) it receives a lot more than 500mm/yr.

Line 95: "the average per capita water resource is only 23% of the Chinese average." Where does this information come from?

I cannot agree that the spatial distribution of the LAI reduction in Fig 11b is consistent with the area of water storage deficit in Fig 6.

The authors consistently refer to the probability of precipitation in the drought being 84%. I don't understand what they mean by this.

Technical corrections:

Many, perhaps most, of the references are improperly or inadequately cited both in the text and in the reference list.

Section 2.2.3: Precipitation data from the Chinese Met. Admin. have been gridded using a 1984 SYMAP system and "extensively verified for runoff, evapotranspiration, and soil moisture (Zhang et al., 2014)." It is not at all clear what this reference to Zhang et al means. Their work is a dataset of hydrological fluxes and states at 3 hr interval for China for the period 1952-2012 developed using gridded data and the VIC model already referred to earlier in this paper. Zhang et al report that the data set is available on the web. So have the authors of this paper used the Zhang et al data, and if so, why

don't they just say so? If they don't use the Zhang et al data, what do they use, and why do they refer to Zhang et al in this way?

Section 2.3.1. Given the focus in this paper on drought, it is somewhat surprising that the SPI is simply selected as the drought measure without discussion or reason. What about the more widely used Palmer Index, or even the drought classification of the Chinese Meteorological Association. The description of the SPI given in this section appears to have little relation to what McKee et al wrote. McKee et al say that: "A drought event for time scale i is defined here as a period in which the SPI is continuously negative and the SPI reaches a value of -1.0 or less. The drought begins when the SPI first falls below zero and ends with the positive value of SPI following a value of -1.0 or less." (no page number available). In this paper the authors state (lines 178-179): "When the time periods are small (1 or 6 months), the SPI frequently fluctuates above and below zero (McKee, 1993)." McKee et al do actually say this but it is in reference to their example of the SPI for Fort Collins. It appears to have no relevance to the way the SPI should be used in this study. Note also that the text above is a direct quote from McKee et al though not identified as such. After all that, Section 2.3.1 doesn't actually tell us how they dealt with the SPI, though there is more on that in the results.

Section 2.3.3. Given the spatial distribution of groundwater gauging stations shown in Fig 1b, they do not provide a reasonable cover of the whole area of interest. This matter is not considered at all here in using the first method for estimating GWC. Large parts of the study area have no data.

Line 235: The usual definition of the hydrological year is that it begins at the month of lowest flow/precipitation. In that case the hydrological year here would start in February. So why has it been arbitrarily started in May?

Brian Finlayson The University of Melbourne

---

## Author Comment (AC1) · 10 Oct 2016

Response to the referee comment for article hess-2016-313

Note: The text in black is the original comments from the referee, and the text in blue, headed with "Reply", is the response from the authors.

This manuscript aims to identifying the water deficit under an extreme drought in North China by using the GRACE data, and comparing with the response of vegetation, towards the implications for the well-known South-to-North Water Diversion (SNWD) project. It is not new to investigating the GRACE-derived water storage changes under

droughts. As the author mentioned that several previous studies have done this with varying focuses, but it remains very interesting in the North China, where the water shortage is aimed to be mitigated by the SNWD. Besides, due to the differences on the drought characteristics such as duration and severity, the detection and variation of GRACE-derived water storage changes may vary from place to place. Thus, this manuscript addressed a good scientific question and may attract interests from the community of hydrology, geodesy, and even the public people. To achieve above purposes, the study derived the total water storage anomalies (TWSA) and groundwater storage anomalies (GWSA) from GRACE time-variable gravity data, and compared it with model simulations and LAI. The method used in this study is generally appropriate and the result is reliable. However, some efforts may still be needed to improve the quality of this manuscript and make it easier to be understood. Reply: We greatly thank the reviewer for the constructive comments. According to the comments and suggestions, we provide more information here and we will improve our manuscript. The detailed point-by-point responses are listed below. Major Comments Comment 1 Section 2.3.2, Evaluation of GRACE TWSC: This manuscript compares the so called net recharge (this name is confusing, actually it is commonly written as dS/dt or TWSC in many papers) derived from GRACE and land surface models. It should be noticed that in Equation (2), the runoff should be the net runoff (i.e. outflow minus inflow). Since the North China is a self-defined region with a shape of rectangle, it is definitely not a closed basin. How is the net runoff calculated? Please give more explanations. Instead of the net recharge, there is another way for evaluation, i.e. using the observations of groundwater storage (GWSA) and surface water storage (SWSA), and model simulations of the soil moisture storage (SMSA), with the equation: TWSA = GWSA + SWSA + SMSA. Reply: The net recharge (i.e., the total water storage change, dS/dt) can be calculated with two approaches. One is the water storage-based approach as mentioned above by the reviewer. This approach depends on multi-source data, including GRACE (for TWSA), groundwater storage measurement (for GWSA), surface water storage measurement (for SMSA) and soil moisture simulation/measurement (for

SMSA). So this approach may induce substantial known/unknown uncertainties in the evaluation of GRACE. The other one is the flux-based approach used in this study (i.e, $\triangle S\_i = P\_i - E\_i - R\_i$). It requires data only from GRACE and land surface modeling. Moreover, the precipitation (P), evapotranspiration (E) and runoff (R) are consistent because they are from the same land surface model (VIC or NOAH). Therefore, we employed the flux-based approach. The runoff (including surface runoff and subsurface runoff) simulated by the land surface models (VIC and NOAH) is for each grid cell at 0.25-degree resolution, and it generally flows through the study area in a period less than one month. The 0.25-degree simulation data were aggregated to the entire area of North China and then used in Equation (2). So it does not matter whether the area is a close basin or not.

Comment 2 Section 4.2, Vegetation response: More deep analysis is needed to figure out the vegetation response under the drought. For example, can the monthly LAI help to interpret the response while compared with the monthly TWSA? Reply: This is a useful suggestion. The TWSA represents the changes including surface water storage, soil moisture and groundwater storage. Soil moisture generally has larger impact on the LAI change than the TWSA does. So we analyzed the correlation between LAI and soil moisture. The result is shown below (Figure 1). The variations of LAI and soil moisture have similar patterns. Both reach the low points in 2009 and their Pearson correlation coefficient is up to 0.74. Please note the state of LAI is impacted not only by soil moisture and but also by human activities (e.g., crop planting). Moreover, the spatial distribution of the LAI reduction in 2009 is consistent with the soil moisture deficit to some degree (Please see Figures 1 and 2 and related discussions in the response to Referee 2). Therefore, the vegetation growth has been substantially restricted during the 2009/10 drought event.

Figure1. Soil moisture and LAI variations during the growth season (May-October) in North China Minor Comments Comment 1 Line 22: should be: is one of the most damaging. . . Reply: We will revise it in the manuscript. Comment 2 Line 27: 'quality

data sets' is confusing. Reply: 'quality data sets' used here means the GRACE data are acceptable for the total water storage detection. We will reword the sentence in the manuscript. Comment 3 Line 111: please point out the GRACE data used is Level-2 or Level-3. Reply: The GRACE data used is Level-2. Comment 4 Line 120: 'total water' should be 'total water storage' Comment 5 Line 121: 'groundwater water change' should be 'groundwater storage change' Comment 6 Line 159: 'groundwater table' is better to be replace with 'groundwater level' Comment 7 Line 199: 'to detect groundwater' should be 'to detect groundwater storage' Comment 8 Line 204: 'groundwater level measured in situ' should be 'in situ measured groundwater level' Reply 3-8: Thanks for these useful suggestions. The manuscript will be improved as suggested. Comment 9 Line 209: the symbol of G S M C W is not typically used in the GRACE hydrology community, I suggest to using GWSA = TWSA – SMSA – SWSA – CWSA. Reply: This expression is acceptable in the GRACE community, but some readers may confuse the multiletter variables. For example, GWSA may be misunderstood as $G \times W \times S \times A$. So we kept the simple symbols of G S M C W. The other referee (i.e., the editor) suggested such simple symbols. Comment 10 Line 222: what is precipitation deficit? I can not understand how is the 14 mm and 47 mm derived. Reply: Precipitation deficit is the difference between the precipitation of a period and the long-term mean. So the precipitation deficit of 14 mm is the total precipitation during 2009/10 minus a long-term (1960-2012) average precipitation in North China. Comment 11 Line 253: 'fluctuations' better to be replaced with 'amplitude'. Comment 12 Line 261: 'departure' is not easy to understand, usually we use 'anomaly' or 'difference'. Reply 11-12: Thanks. The manuscript will be revised as suggested. Comment 13 Line 262: it is hard to say the drought events mainly occur in the south of North China, as the groundwater exploitation is complicated in space. Reply: We agree that the groundwater exploitation is complicated in space. Please note drought occurrence generally means more groundwater exploitation. The total water storage is especially low in the south of North China during 2009/10, comparing to other areas where anomalies above zero. After the drought event, the total water storage

recovers to some degree. So the drought impact is severer in the south of North China. Comment 14 Line 264: 'normal' or 'average'? Reply: Both words are right here, while I think the 'normal' to modify a condition or state is better. Comment 15 Line 275: 'public supply' should be 'domestic use' Comment 16 Line 283: 'downward' should be 'decreasing' Comment 17 Line 287: delete 'approximately', same in many other places throughout the manuscript Comment 18 Line 311: 'groundwater decline' should be 'groundwater level decline' Comment 19 Line 352-354: references are not commonly found in the conclusion section Reply 15-20: Thanks for the valuable suggestions. Comment 20 Figure 1: the year for annual precipitation, long-term mean or some specific year? Reply: It is the long-term mean annual precipitation. Comment 21 Figure 1: 'Groundwater Gauge Stations' should be 'Groundwater Level Monitoring Wells' Reply 21: Will be revised as suggested. Thanks. Comment 22 Figure 2(b): do not use abbreviation for the name of y-axis Reply: We will replace it as difference. Comment 23 Figure 2(a): what is the time for annual average, 53-year mean? Reply: Yes, the annual average is the mean from 1960 to 2012. Comment 24 Figure 3(b): No name for the x-axis in the small figure inside. The green histogram and red dots seems represent the same thing. If not, please give more explanation. Reply: The histogram shows each year's precipitation ordered from high to low, while red dots represent each year's probability using Weibull equation (Helsel D, 2002). Comment 25 Figure 5: What is the meaning of '/' in 'May/Jun 2009' and the others? Is it the average TWSA of May and June 2009? Please make it clearer. Reply: Yes, 'May/Jun 2009' is the average total water storage anomaly of May and June 2009. Comment 26 Figure 7: The name of the right y-axis should be 'Equivalent Water Height (mm)' Reply 26: Thanks for your valuable suggestions. ReferenceïijŽ Helsel D, H. R.: Statistical Methods in Water Resources Techniques of Water Resources Investigations, U.S. Geological Survey, chapter A3 of Book 4, 2002.

Please also note the supplement to this comment:
http://www.hydrol-earth-syst-sci-discuss.net/hess-2016-313/hess-2016-313-AC1-

supplement.pdf

---

## Author Comment (AC2) · 10 Oct 2016

Response to the referee comment for article hess-2016-313

Note: The text in black is the original comments from the referee, and the text in blue, headed with "Reply", is the response from the authors. General Comments Comment 1 The focus on drought alleviation in relation to the SNWD seems inappropriate. Rather the SNWD was developed to deal with the high water demand in North China which is a problem irrespective of whether drought conditions occur. Further, over-extraction of groundwater from many areas of North China, especially the North China Plain, has a

much greater and more permanent impact on groundwater storage than droughts. To ascribe North China's problems with water as being caused by droughts is an extreme simplification of the water resources situation there. Reply: We would like to thank the referee, Dr. Finlayson, for providing useful comments and suggestions. We are sorry that some expressions in the manuscript make the referee misunderstand the focus of this study. We do not attempt to attribute the water problem in North China to droughts. Sure, the SNWD was developed to relax the water shortage in North China and to facilitate groundwater recovery. However, the stability and efficiency of the SNWD are significantly impacted by climate change, especially the extreme weather and climate events (e.g., drought and flood events). So climate change brings substantial challenge to the operation of the SNWD. The primary objective of this study is to identify the recent typical drought event during 2009/10 in North China using GRACE data. Quantifying water deficit and persistence for this drought is expected to provide implication for the implementation of the SNWD project (e.g., the timing and the volume for water transfer), although this is not the main focus of our study. So we roughly evaluated whether the water transferred by the project can relieve the water deficit at this level of drought. We will improve the expressions regarding the focus to avoid misunderstanding. Comment 2 The introduction is rather divorced from the content of the body of the paper which focuses on the use of GRACE in analyzing water storage in the NCP region. The introduction should focus on the main topic of the paper with a review of key papers from which it can be shown that the work reported in this paper is a worthwhile development on what has been already done in this field and then go on to show how this will be achieved. Reply: Thanks for the kind suggestion. We will add more information about the water resource condition in North China and briefly review GRACE application with drought detection. Comment 3 The Data and Methods Section (Section 2) lacks a clear description of the methodology of this study. It begins with a rather misleading description of the field area and describes the GRACE system and data sources, proceeding to say that the hydrological modeling described is taken from some other source and is not part of this research. The modeling is reported to have

been "evaluated in North China with acceptable uncertainties" (lines 147-148) but with no description of what this means. Reply: Although GRACE data have been widely used to detect water storage condition, the data need validation for a specific region of interest due to their uncertainties. Validation of GRACE data is essential for remote sense data (Wang et al., 2014;Syed et al., 2008). So we employed simulation data from two land surface models (i.e., VIC and NOAH) and compare the net recharge calculated from GRACE and the simulations. Moreover, GRACE data require simulation data to isolate groundwater storage (equation (4)). At the beginning of section 2.2.2, we stated that "to validate the terrestrial water storage measurements of GRACE, water fluxes (i.e., runoff and evapotranspiration) and soil moisture from two land surface models were used in this study". We did not perform model simulations and evaluations. Instead, we took data from other studies in which the simulation data have been well evaluated.

Comment 4 Water resources are heavily used in this region, especially in the southern part on the North China Plain, so the focus on drought and failure to take more account of resources use is strange. There is some discussion of this in the section from line 305 to 312, beginning: "One may wonder the role of human over-use of the water resources." Indeed. Reply: We agree that heavy use and over-extraction of water resources are interesting topics and there are lots of studies. Our study focused on the typical drought event, identifying the drought cycle and the water deficit. Human activities have significant influence on water storage change. Figure 10 shows that groundwater withdraw continuously decreased during past decade. Therefore, the water deficit in 2009/10 is dominated by the drought event. We will provide more discussions on the interaction of the drought and water use in North China.

Specific Comments Comment 1 This paper begins with an incorrect statement by claiming that a paper by Palmer 2002 is the source for a statement that "The global climate system has significantly changed in recent years, leading to an increased frequency of extreme weather and other disaster events". The paper is actually by Palmer

and Räisänen and it does not say that there has been a change in recent years leading to increased frequency of extreme events but rather it discusses the probability of this occurring in the future. This paper cannot be used to support such a claim for north China. Also, the paper was written 14 years ago and a lot more has been published on this topic since. I wonder how many other references used in this paper would stand up to scrutiny? Reply: Thanks for correcting the inappropriate citation. We will check all citations and references. Actually, a few studies indicated that the frequency of extreme weather and climate events has increased during past decades (Leng et al., 2015;Qin et al., 2015).

Comment 2 Lines 49-50: "Drought frequently occurs in most areas of China and accounts for 35% of all economic losses from disasters." No source is given for this specific piece of information. Reply: The information is from the book 'Song, L. C., Z. Y. Deng, and A. X. Dong (2003), Drought, China Meteorol. Press, Beijing (in Chinese) '. Similar statement was also given in a few papers (Ye et al., 2012;Gao and Yang, 2009) . Comment 3 Line 78: 'SPI' not defined until line 167. ReplyïijŽIt is abbreviated from the Standardized Precipitation Index. We will make the definition before it appears. Comment 4 North China, as defined in Fig 1b, is not the area with the most severe water shortage in China (line 50); only the western part of that area is semi-arid; and based on the authors' own map of precipitation distribution (Fig 1a) it receives a lot more than 500mm/yr. ReplyïijŽWhether the area experiences water shortage depends not only on the water input (e.g., the precipitation), but also the water demand. North China receives precipitation more than 500 mm, but its population density is over 500 person/km2. Particularly, the population density exceeds 1000 person/km2 in the eastern part of North China. With the high population density, rapid increase of water demand for agriculture and industry necessarily induces the shortage of water resources. Moreover, the increasing frequency of drought has exacerbated the situation of water availability. Comment 5 Line 95: "the average per capita water resource is only 23% of the Chinese average." Where does this information come from? Reply: Water shortage is most serious in North China, although the rate of the average per

capita water resource between North China and the whole of China varies according to different studies (Jun, 2010;Li and Mu, 2006). Comment 6 I cannot agree that the spatial distribution of the LAI reduction in Fig 11b is consistent with the area of water storage deficit in Fig 6. ReplyïijŽPlant growth is more sensitive to soil moisture storage than the total water storage generally. So we analyzed the correlation of LAI with soil moisture in space and time. The relative reductions of LAI and soil moisture were calculated by Red=(V_i-V_c)/V_c (1) Where Red is the relative reduction of LAI (or soil moisture), Vi and Vc are the average in 2009 and the climatology value for LAI (or soil moisture), respectively. As shown in Figure 1 and 2, LAI and soil moisture have similar distribution in space. Both of them show significant reductions in the northwest of North China (Part A), but have minor increases in Part B. Their distributions are not so consistent in the Southeast (Part C) where there are intensive human activities (e.g., crop cultivation and groundwater extraction). By and large, LAI and soil moisture storage show great reductions in the 2009/10 drought event. Figure 3 presents the time series of the LAI and soil moisture during the plant growth season (May-October). We can see that the change of LAI agrees well with soil moisture in time, and their Pearson correlation coefficient is up to 0.74. Moreover, both of them reach the low points in 2009. Therefore, the LAI reduction is consistent with the distribution of soil moisture to some degree in space and time. Vegetation growth is substantially constrained during the drought.

Figure 1. Soil Moisture Deficits in 2009 Relative to the Average Soil Moisture Conditions.

Figure 2. LAI Reduction in 2009 Relative to the Average Conditions.

Figure 3. Soil moisture and LAI during the plant growth season (May-October) in North China Comment 7 The authors consistently refer to the probability of precipitation in the drought being 84%. I don't understand what they mean by this. ReplyïijŽAs mentioned in the manuscript in line 239, the percent of 84% is a statistic which means the probability of precipitation in 2009 during the 53 years (1960 to 2012), indicating a severely dry

episode in 2009. We used the Weibull equation (Helsel D, 2002), p=n/(m+1)×100%, where n is the order of the yearly precipitation (n = 43 for the precipitation in 2009) and m is total number of years ( m = 53), then we got p = 84% for the precipitation in 2009. Technical Corrections Comment 1 Section 2.2.3: Precipitation data from the Chinese Met. Admin. have been gridded using a 1984 SYMAP system and "extensively verified for runoff, evapotranspiration, and soil moisture (Zhang et al., 2014)." It is not at all clear what this reference to Zhang et al means. Their work is a dataset of hydrological fluxes and states at 3 hr interval for China for the period 1952-2012 developed using gridded data and the VIC model already referred to earlier in this paper. Zhang et al report that the data set is available on the web. So have the authors of this paper used the Zhang et al data, and if so, why don't they just say so? If they don't use the Zhang et al data, what do they use, and why do they refer to Zhang et al in this way? Reply: Sorry for that the data description may confuse the referee. The precipitation data used in this study were from Zhang et al (2014). The simulation data of runoff, evapotranspiration and soil moisture were from two sources: VIC simulation by Zhang et al (2014) and NOAH simulation in Global Land Data Assimilation System (GLDAS). Comment 2 Section 2.3.1. Given the focus in this paper on drought, it is somewhat surprising that the SPI is simply selected as the drought measure without discussion or reason. What about the more widely used Palmer Index, or even the drought classification of the Chinese Meteorological Association. The description of the SPI given in this section appears to have little relation to what McKee et al wrote. McKee et al say that: "A drought event for time scale i is defined here as a period in which the SPI is continuously negative and the SPI reaches a value of -1.0 or less. The drought begins when the SPI first falls below zero and ends with the positive value of SPI following a value of -1.0 or less." (no page number available). In this paper the authors state (lines 178-179): "When the time periods are small (1 or 6 months), the SPI frequently fluctuates above and below zero (McKee, 1993)." McKee et al do actually say this but it is in reference to their example of the SPI for Fort Collins. It appears to have no relevance to the way the SPI should be used in this study. Note also that the text above is a direct quote from McKee

et al though not identified as such. After all that, Section 2.3.1 doesn't actually tell us how they dealt with the SPI, though there is more on that in the results. Reply: The use of the SPI as the drought measure is based on two reasons. First, SPI is easily calculated and it requires precipitation data only. Second, SPI has its advantage of characterizing multi-scale atmospheric drought situation (e.g., 1-month, 6-month and 12-month). This multi-scale representation is suitable to detect different water storage responses. For example, soil moisture storage has a short-term response (< 1 month) to the precipitation anomaly while groundwater shows a long-term response (> 6 months). So we employed the SPI in this study to identify the precipitation deficit and then discuss the changes of total water storage and groundwater storage. Certainly, the Palmer Index (generally, the Palmer Drought Severity Index, PDSI) is also widely used and its calculation depends on precipitation and temperature. It has been most effective in determining long-term drought. Although the statement by McKee et al (1993) is with respect to a case study, we argue it is relevant for general situations. The SPI values generally move frequently around zero for small scales (e.g. 1 month), because it represents the fluctuation of monthly precipitation. Figure 3a in this study also shows this pattern. Comment 3 Section 2.3.3. Given the spatial distribution of groundwater gauging stations shown in Fig 1b, they do not provide a reasonable cover of the whole area of interest. This matter is not considered at all here in using the first method for estimating GWC. Large parts of the study area have no data. Reply: We admit that the ground measurements of groundwater table do not adequate cover the whole area of interest due to data availability. Based on such data set, the calculation of groundwater storage condition (the first method) may have substantial uncertainties. But the measurements are used as auxiliary data to compare with the GRACE which is primarily applied to detect the changes of water storages. This method and the data have also been used in a few studies (Huang et al., 2015;Feng et al., 2013). Comment 4 Line 235: The usual definition of the hydrological year is that it begins at the month of lowest flow/precipitation. In that case the hydrological year here would start in February. So why has it been arbitrarily started in May? Reply: Vegetation in North

China generally begins to grow in May because of favorable air temperature and precipitation. In this study, we attempted to discuss the drought impact on the vegetation growth. So the hydrological year is defined in May-April. Similar results can also be found in Barriopedro et al. (2012).

Referencesïjž Barriopedro, D., Gouveia, C. M., Trigo, R. M., and Wang, L.: The 2009/10 Drought in China: Possible Causes and Impacts on Vegetation, Journal of Hydrometeorology, 13, 1251-1267, 10.1175/jhm-d-11-074.1, 2012. Feng, W., Zhong, M., Lemoine, J.-M., Biancale, R., Hsu, H.-T., and Xia, J.: Evaluation of groundwater depletion in North China using the Gravity Recovery and Climate Experiment (GRACE) data and ground-based measurements, Water Resources Research, 49, 2110-2118, 10.1002/wrcr.20192, 2013. Gao, H., and Yang, S.: A severe drought event in northern China in winter 2008–2009 and the possible influences of La Niña and Tibetan Plateau, Journal of Geophysical Research, 114, 10.1029/2009jd012430, 2009. Helsel D, H. R.: Statistical Methods in Water Resources Techniques of Water Resources Investigations, U.S. Geological Survey, chapter A3 of Book 4, 2002. Huang, Z., Pan, Y., Gong, H., Yeh, P. J. F., Li, X., Zhou, D., and Zhao, W.: Subregional-scale groundwater depletion detected by GRACE for both shallow and deep aquifers in North China Plain, Geophysical Research Letters, 42, 1791-1799, 10.1002/2014gl062498, 2015. Jun, X.: Water Security and Ground Water Problem to Changing Environment in North China, International Symposium on Groundwater Sustainability (ISGWAS), 14, 2010. Leng, G., Tang, Q., and Rayburg, S.: Climate change impacts on meteorological, agricultural and hydrological droughts in China, Global and Planetary Change, 126, 23-34, 10.1016/j.gloplacha.2015.01.003, 2015. Li, R., and Mu, X.: Challenges and Conservation Measures: Water Resources of the North China Plain, Water for Irrigated Agriculture and the Environment, 6, 2006. Qin, Y., Yang, D., Lei, H., Xu, K., and Xu, X.: Comparative analysis of drought based on precipitation and soil moisture indices in Haihe basin of North China during the period of 1960–2010, Journal of Hydrology, 526, 55-67, 10.1016/j.jhydrol.2014.09.068, 2015. Syed, T. H., Famiglietti, J. S., Rodell, M., Chen, J., and Wilson, C. R.: Analysis of terrestrial water storage changes from

[Figure]

GRACE and GLDAS, Water Resources Research, 44, 10.1029/2006wr005779, 2008. Wang, H., Guan, H., Gutiérrez-Jurado, H. A., and Simmons, C. T.: Examination of water budget using satellite products over Australia, Journal of Hydrology, 511, 546-554, 10.1016/j.jhydrol.2014.01.076, 2014. Ye, T., Shi, P., Wang, J. a., Liu, L., Fan, Y., and Hu, J.: China's drought disaster risk management: Perspective of severe droughts in 2009–2010, International Journal of Disaster Risk Science, 3, 84-97, 10.1007/s13753-012-0009-z, 2012.

Please also note the supplement to this comment:
http://www.hydrol-earth-syst-sci-discuss.net/hess-2016-313/hess-2016-313-AC2-supplement.pdf

---

## Editor Comment (EC1) · J. Seibert (Editor) · 12 Oct 2016

Both reviewers have major concerns with the manuscript and raise a list of issues ranging from needs to clarify the presentation to more substantial questions on the used approaches. Altogether, addressing these issues will require substantial major revisions. The methods and used data were rather unclear, which means that the reviewers only partly were able to assess what exactly has been done. The revised manuscript will have to address all these comments. The replies by the authors where difficult to read due to the lack of paragraphs. As far as I can see the authors provide

some way forward to address the reviewers' concerns, although some replies, such as to reviewer 2 comment 3, were not fully convincing. Here certainly more efforts by the authors are needed.

As a small but important additional note: in figure 1 islands south of China are included. These are disputed borders and as there are from the content of the manuscript/figure no need to include these islands, I'd ask you to remove the inset in the map of figure 1. (see http://www.nature.com/nature/journal/v478/n7369/pdf/478285a.pdf)

Best regards,

Jan Seibert
* * *